# Urban Residents' Preferred Walking Street Setting and Environmental Factors: The Case of Chengdu City

**Qian Yan** [1] [iD], **Shixian Luo** [2],* [iD] **and Jiayi Jiang** [3,4]

1    School of Landscape Architecture, Beijing Forestry University, Beijing 100083, China; yanqian603@126.com
2    School of Architecture, Southwest Jiaotong University, Chengdu 611756, China
3    School of Architecture, Soochow University, Suzhou 215123, China; jyjiang@suda.edu.cn
4    China-Portugal Joint Laboratory of Cultural Heritage Conservations Science Supported by the Belt and Road Initiative, Suzhou 215123, China
*    Correspondence: shixianluo@yahoo.com; Tel.: +86-18108082672

**Abstract:** To date, most studies on building environments and walking behavior have utilized top-down approaches (e.g., big data or social media data) yet lack bottom-up approaches to verify their findings. Therefore, this study divided urban streets into three main settings (community streets, waterfront paths, and urban greenways) and collected data from a sample of 411 urban residents in Chengdu via an online questionnaire to examine the impact of street environmental factors on their choice of walking path. It was found that: (1) people with higher levels of education preferred streets with water bodies as walking paths; (2) the environmental quality of the physical and aesthetic aspects both had an impact on residents' choices, and the aesthetic environmental quality had a stronger impact; (3) the impact of most infrastructures on community streets was stronger than on other streets; (4) residents were more concerned about the environmental quality of waterfront paths and urban greenways. Based on these findings, three design patterns for residents' preferred street environments are proposed.

**Keywords:** walking streets; build environments; environmental factors; physical environmental factors; aesthetic environmental factors; preferred path; street design; online questionnaire; bottom-up approach

## 1. Introduction

### 1.1. Importance of the Built Environment

By 2030, the urban population will have grown due to rapid urbanization. The influx of a large number of people into urban areas has resulted in worsening congestion and substandard living conditions. Owing to the high population density and lack of public facilities, residents in cities could experience various mental and physical health issues. Hence, addressing the health concerns of urban residents has become a global challenge for cities [1].

The built environment, as the most common and active area for residents [2,3], is defined as part of the physical environment that is constructed by human activity [4]. Therefore, it is considered to significantly influence the travel habits of urban residents [2,3]. In addition to providing space for simple daily travel, the built environment accommodates a variety of social activities [5]. The overall attractiveness and safety of the built environment encourage residents to be more physically active, thereby increasing opportunities for physical activity and social interaction. Accordingly, well-designed urban built environments can reduce the risk of disease among residents. A comprehensive definition of health is "health is a state of complete physical, mental and social well-being and not merely the absence of disease or infirmity" [6]. In addition, a pleasantly built environment may affect pedestrian behavior by increasing the frequency of outdoor activities, which has positive psychological and physical health effects [7]. Consequently, as an essential component of cities, a well-designed built environment can enhance the residents' quality of life.

Furthermore, the built environment, especially the neighborhood street environment, plays an important role in influencing the health of urban residents. Studies at the neighborhood level have observed the subjective impacts of streets on residents. In addition to the impact of the built environment on residents' walking preferences for basic travel, the street environment can also influence residents' outdoor recreational behaviors, such as walking [8,9]. The relationship between the built environment and walking behavior has been the subject of numerous studies in recent years [10–13].

### 1.2. Value of Walking Behavior for Public Health

Walking is one of the most accessible forms of physical activity [14] and can readily contribute to the population's physical health. Through a review of prior research, walking can reduce the likelihood of cardiovascular disease, certain cancers, type 2 diabetes, osteoarthritis, and osteoporosis, the risk of death from cardiovascular disease and cancer [15], injury, improves bone strength, accelerates metabolism, controls blood pressure, and enhances the mental health of pedestrians [16–18]. Walking can also maintain long-term weight loss as well as increase social confidence [19,20].

Additionally, walking has a positive effect on the mental health of a population. It can reduce blood pressure, symptoms of depression and anxiety [21], and perceived stress. It can also improve cognitive health [22], concentration, and social cohesion [23]. Consequently, walking can improve both the physical and mental health of residents, making the enhancement of urban public health extremely valuable.

### 1.3. Effect of Street Environmental Quality on Walking Behavior

A well-designed street can enhance residents' quality of life and encourage walking, thereby improving their physical and mental health. Measuring the quality of a street and designing one that is aesthetically pleasing has become a significant area of study. In general, the quality of an environment can be described in terms of objective physical environmental quality and subjective aesthetic environmental quality.

The physical environmental quality of a street (such as infrastructure and walking conditions) influences the probability that residents will utilize it as a pedestrian space. Previous research has indicated that the provision of infrastructure increases street walkability [24–26]. When determining walking route options, infrastructure data frequently include parks, restaurants, cafes, stores, medical services, and recreational facilities [27]. In addition, the walking conditions of streets affect residents' walking, and previous studies have found that diverse neighborhoods well-served by public transportation have higher levels of walking activity [28–30]. Sidewalk width also influences pedestrian path preferences, while the number of roadway intersections, the number of rising steps, and the presence of crosswalks or overpasses impact residents' walking preferences [31–34]. Moreover, the environmental quality of the street can also impact residents' walking preferences, and the presence of street items such as fountains, public art installations, public furniture, and green spaces can increase the appeal of a street to pedestrians [35–37].

The aesthetic quality of streets has long been discussed in terms of the nature of beauty, which Baumgarten formalized in the 18th century under the term "aesthetics". Around this time, people began philosophizing about the human sense of beauty. Modern design is not only concerned with aesthetic qualities but also with aesthetic perception [38]. Some studies have found a positive correlation between walking and perceptions of attractiveness, aesthetics, and greenery [39–41]. Street aesthetics shape urban design qualities (such as imageability, closure, human scale, and transparency), which affect walkability by eliciting individual responses (such as feelings of safety, comfort, and interest) [42]. In addition, Koo et al. (2022) examined the relationships between harmony, rhythm, balance, order, complexity, scale, maintenance, and pedestrians on streets [43]. Zhao et al. (2018) discovered that the soundscape and olfactory characteristics of the environment influence pedestrian perception [44–47]. Accordingly, this study aimed to investigate and

discuss the reasons why urban residents choose a particular street for walking based on two dimensions: physical quality and aesthetic quality.

*1.4. Research Questions*

Alan Jacobs was the first to propose a path survey methodology for pedestrian streets. Later, Montello et al. gathered data on urban streets and pedestrian perceptions via field surveys, interviews, and questionnaires [48]. Gradually, quantitative methods, such as stated preference surveys, have been incorporated into field surveys, frequently employing manually collected small data sets. Despite numerous studies addressing the influence of the street environment on residential walking activity, the majority of studies on path behavior preferences have relied on top-down analyses that combine path information derived from large data sets with an urban context [49,50]. It is essential to note that top-down studies are typically valid; however, validating their accuracy is difficult because most conclusions are based on speculation from numerical results. More bottom-up studies (such as questionnaires) are required to directly validate the influence of each environmental element on path selection.

In addition, the concept of "street" has been rather vague in many recent studies. Zhong et al. (2022) identified three types of streets: "urban sidewalks," "community sidewalks," and "green space sidewalks," and discovered that the three distinct street environments had different impacts on people and that various environmental elements have different positive and negative impacts on residents [51]. Hence, to apply these findings to specific street types, as in Zhang et al. (2022), we first categorized urban streets into three types: community streets (built environment), waterfront paths (built environment with water elements), and urban greenways (built environment with green elements) [51]. Using a questionnaire, this study attempted to answer the following key questions:

1. What kind of street setting do different people prefer as a walking path?
2. Which physical characteristics influence the selection of preferred walking streets based on the street environment?
3. Depending on the street environment, which aesthetic qualities influence residents' choice of preferred walking streets?

## 2. Materials and Methods

### 2.1. Research Area

This study was conducted in Chengdu, Sichuan Province, China (Figure 1). The urban area has a population of 9.52 million (2019). Chengdu adopted the "Internationalized Community Construction Plan (2018–2022)" and "Internationalized Community Construction Policy Measures" (http://www.cdswszw.gov.cn/zcfg/Detail.aspx?id=5930, accessed on 29 May 2019), with the objective of creating a high-quality, harmonious, and livable community, aiming to create more comfortable and convenient urban street spaces for residents of Chengdu's central city. The Chengdu Municipal Government proposed the "Livable Chengdu" concept in 2022, aiming to improve the "walkability" of urban streets and encourage more residents to use them for outdoor recreation. As a result, a design strategy to improve city streets has become one of the city's most important policies.

### 2.2. Questionnaire Design

The authors designed a series of questions regarding walking in urban streets and collected data from a sample of urban residents. Owing to the cost of the study and the limitations of the COVID-19 pandemic, we collected data via online survey platforms and social media.

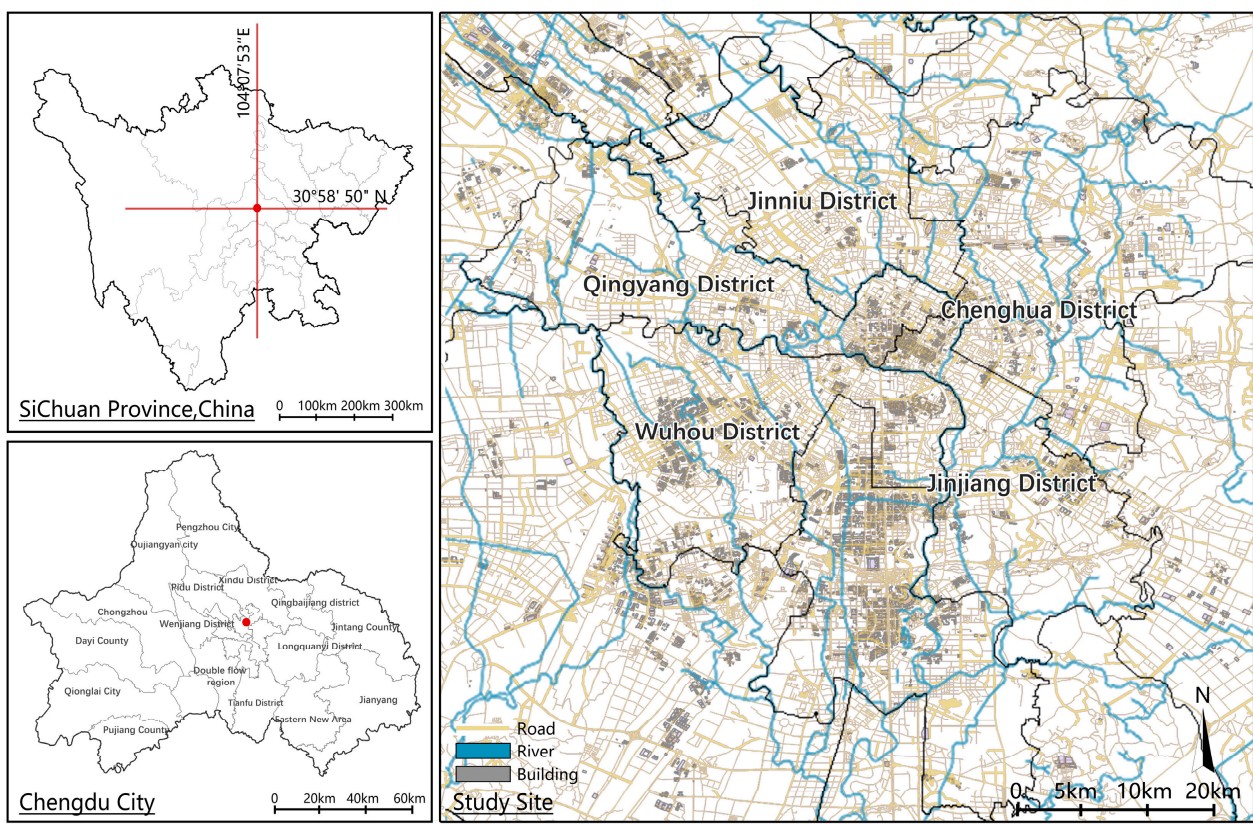

**Figure 1.** The location of the study area.

The online questionnaire consisted of three sections. The first section collected sociodemographic information from the residents, including their gender, age, level of education, and occupation. The second section categorized urban streets into three settings (community streets, waterfront paths, and urban greenways), and respondents were asked to choose a certain street setting as their daily walking environment. In the third section, we presented 46 factors compiled from the previous literature (Table 1) that may influence residents' choice of streets for walking, considering the knowledge of urban planning and landscape architecture. Residents were asked to indicate how each item influenced their choice of walking in a particular street setting on a scale from −3 to +3. Specifically, based on the street settings selected in the second section, −3 indicated a strong negative impact, where the street would not be chosen as a walking path due to the presence of the item, and +3 indicated a strong positive impact, where the street would definitely be chosen as a walking path due to the presence of the item. Figure 2 depicts the basic structure of the questionnaire. Please see Appendix A for the complete questionnaire.

**Table 1.** The list of all the independent variable factors for this study and the rationale for their selection.

| Environmental Factors | Environmental Factor Layer 1 | Environmental Factor Layer 2 | Rationale |
|---|---|---|---|
| Physical Environmental Factors | Infrastructure (road functional facilities) | Lighting<br>Guardrails<br>Signage<br>Garbage cans<br>Power poles<br>Billboards<br>Street cameras<br>High-voltage boxes | Streetscape elements such as guardrails, trash cans, and other streetscape elements are related to pedestrian walking preferences [52]; urban furniture is related to elements of the built environment and the pedestrian walking experience [53]. |

**Table 1.** *Cont.*

| Environmental Factors | Environmental Factor Layer 1 | Environmental Factor Layer 2 | Rationale |
|---|---|---|---|
| Physical Environmental Factors | Infrastructure (service facilities) | Retail stores<br>Restaurants<br>Teahouses<br>Bars<br>Cafes<br>Internet cafes<br>Hypermarkets<br>Mobile stalls<br>Toilets<br>Community clinics<br>Community fitness equipment<br>Community service centers | Changes in a built environment's general functions are significant, and pedestrians prefer routes that pass through more stores and services on the ground floor [31,54]. Providing public amenities can boost walkability [24–26]. When determining pedestrian route selection, public amenity data frequently include parks, restaurants, cafes, stores, medical services, and recreational facilities [27].<br>Some pedestrians' walking preferences are positively correlated with the presence of restrooms [55]. |
| | Infrastructure (road traffic facilities) | Bus Stations<br>Subway Stations<br>Cab stops<br>Bicycle parking | The volume of motor vehicle traffic also influences pedestrian preferences [56]. The presence of subway station entrances and exits enhances the vitality of a street. Additionally, bus stops, transit stations, and taxi stand influence residents' walking patterns [45,57]. |
| | Pedestrian conditions | Traffic lights<br>Crossroads<br>Ramps<br>Sidewalks<br>Pedestrian bridges<br>Underpasses<br>Level ground<br>Ground paving<br>Road greening<br>Green space along The street<br>Artificial landscape<br>Litter on the street<br>Seating<br>Gazebo<br>Fountain<br>Street tree box with seating | High activity levels near underpass entrances and pedestrian bridges also influence residents' walking activity [57]; pedestrian path preferences are associated with sidewalks [53]. Green space behavior is positively correlated with pedestrian walkability [53]; pedestrian walking and pocket parks are positively correlated [55]; benches are also associated with pedestrian walking [52,53]. |
| Aesthetic environmental factors | Environmental quality | Diversity<br>Facility Accessibility Uniformity<br>Novelty<br>Maintainability<br>Charm Sense of mystery | Diversity can be used to evaluate the impact of the environment on its inhabitants [58,59]; coherence has many characteristics, including a reflection of unity, balance, harmony, direction, and legibility, as well as understanding the totality of place and its relationship to itself [59–61]; mystery is associated with perceived complexity and attractiveness [58]. |
| | Subjective perception | Pleasant sounds<br>Pleasant odors | The multisensory nature of aesthetic experience is connected with the sounds of the built environment that impact pedestrians [58,62,63], the richness and diversity of odor characteristics, etc., which reflect the observed diversity of things [58,59,64]. |

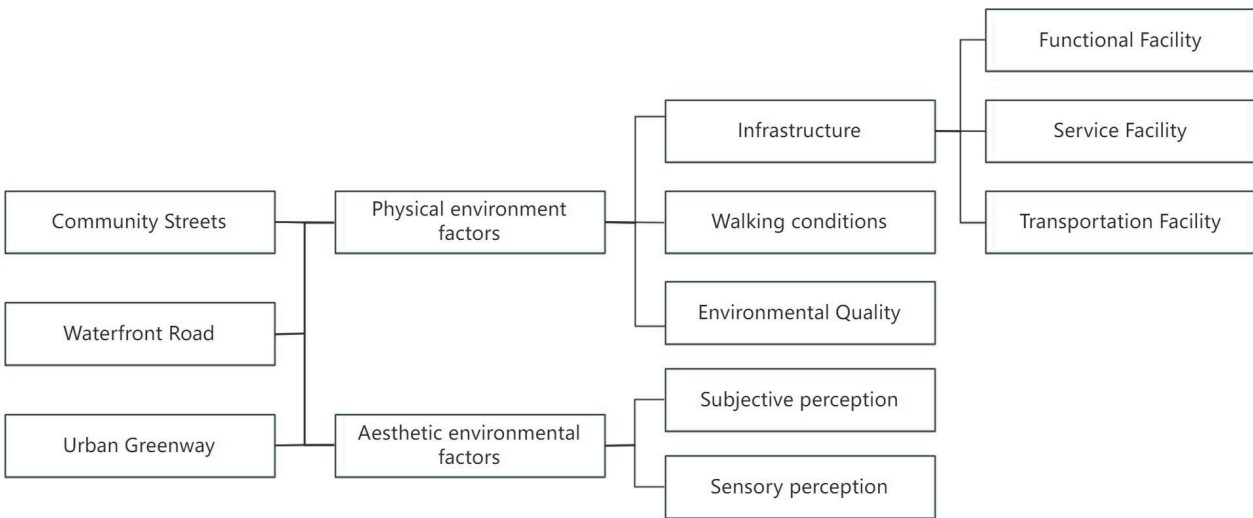

**Figure 2.** The basic framework of the online questionnaire.

### 2.3. Data Collection and Participants

Using the Wenjuanxing platform (https://www.wjx.cn/, accessed on 30 March 2023), a web-based questionnaire was distributed to residents in the Chengdu metropolitan area. In addition, the authors posted a link to the questionnaire on their social media accounts and encouraged their friends and family to complete it. Data were collected from 25 September to 20 October 2022 from people who had resided in downtown Chengdu for more than three years. The questionnaire could be completed at any time throughout the day (including weekdays and weekends) by logging in. To ensure that the questionnaires were from the study area, only data from Chengdu were selected. Questionnaires with response times of less than one minute and incomplete questionnaires were excluded. Finally, 411 valid questionnaires were collected from a wide range of age groups and professions, with the participants taking an average of 9 min to complete the questionnaire.

### 2.4. Data Analysis

Before distributing the questionnaires, a minimum number of required responses was determined using an online sample-size calculator (https://www.surveysystem.com/sscalc.html, accessed on 30 March 2023). Substituting the Chengdu metropolitan area's population of 9.52 million, we established a 95% confidence level and a 5% margin of error for the calculation, and the results indicated that a minimum sample size of 384 was required, so the current number of questionnaires was sufficient.

The final statistical analysis utilized 411 questionnaire responses, and Microsoft Excel was used to manage experimental data. The response frequencies and percentages for each population group were compiled and presented separately. The frequencies of residents' preferred walking street settings (community streets, waterfront paths, and urban greenways) were tallied separately. Multinomial logistic regression analysis was used to explore how respondents' sociodemographic characteristics influenced the choice of their preferred street setting. Moreover, reliability and validity tests of the questionnaire were conducted, as indicated by Cronbach's α coefficient and the Kaiser–Meyer–Olkin statistic, respectively. Subsequently, the means of the environmental factors (both physical and aesthetic) were calculated for each of the three walking street settings separately. Meanwhile, the Kruskal–Wallis H test (with post hoc comparisons) was used to analyze whether there was a difference in the response of 49 environmental factors between the three street settings. Subsequently, given that the scale ranged from −3 to +3, we labeled each factor according to one of the three levels of impact and based on the absolute value of the mean: almost no impact ($|Mean| < 1$), moderate impact ($2 > |Mean| > 1$), and strong impact ($|Mean| > 2$). No impact indicated that the public had no uniform tendency or

perceived little impact from this factor; therefore, it was not a design priority and could be ignored. A moderate impact meant that residents believed that the factor had a medium impact; therefore, these factors should not be disregarded. A strong impact indicated that most respondents concurred that the factor had a significant impact; consequently, these factors should be prioritized in the design. Finally, based on the findings, three design recommendations could be made for street environments. All statistical analyses were performed using Statistical Package for the Social Sciences (SPSS; version 20.0; SPSS Inc., Chicago, IL, USA), and the level of significance was set at $p < 0.05$.

## 3. Results

### 3.1. Basic Information

In total, 411 residents were included in this study. Table 2 displays the profiles of the respondents who participated with regard to gender, age, education level, occupation, driver's license status, and monthly income. According to the results, the percentage of female respondents was slightly higher than that of male respondents (58.64% to 41.36%). In addition, the number of respondents was similar in all three age groups (18–25, 26–55, and over 55 years old). The majority (40.63%, $N = 167$) had a bachelor's degree. The rest were classified as unemployed/retired (24.82%, $N = 102$), students (22.14%, $N = 91$), or corporate employees (16.55%, $N = 68$), which comprised the largest proportion of respondents. In addition, the majority had a driver's license (71.53%, $N = 294$) and earned between 4500 RMB and 8000 RMB (32.36%, $N = 133$). Moreover, Figure 3 shows the frequencies of residents' preferred walking street settings separately: community streets, $N = 57$; waterfront paths, $N = 152$; and urban greenways, $N = 202$.

**Table 2.** Profile of the respondents.

| Category | Sub-Category | Frequency ($N = 411$) | Percentage of Respondents (%) |
|---|---|---|---|
| Gender | Female | 241 | 58.64 |
| | Male | 170 | 41.36 |
| Age | 18–25 years old | 107 | 26.03 |
| | 26–55 years old | 145 | 35.28 |
| | 55+ years old | 159 | 38.69 |
| Educational Level | High School and below | 86 | 20.92 |
| | College | 86 | 20.92 |
| | Undergraduate | 167 | 40.63 |
| | Graduate | 72 | 17.52 |
| Occupation | Medical personnel (doctors, nurses) | 14 | 3.41 |
| | Teachers, lawyers, service industry workers (caterers/drivers/salesmen, etc.) | 35 | 8.52 |
| | Freelancer (e.g., writer/artist/photographer/guide, etc.) | 34 | 8.27 |
| | Workers (e.g., factory workers/construction workers/city sanitation workers, etc.) | 16 | 3.89 |
| | Company employees | 68 | 16.55 |
| | Career/civil servants/government workers | 32 | 7.79 |
| | Students | 91 | 22.14 |
| | Housewife | 19 | 4.62 |
| | No job/retired | 102 | 24.82 |
| Have a driver's license | Yes | 294 | 71.53 |
| | No | 117 | 28.47 |
| Monthly income | Less than 1500 RMB | 57 | 13.87 |
| | 1500–3500 RMB | 82 | 19.95 |
| | 3500–4500 RMB | 67 | 16.3 |
| | 4500–8000 RMB | 133 | 32.36 |
| | 8000–15,000 RMB | 55 | 13.38 |
| | More than 15,000 RMB | 17 | 4.14 |

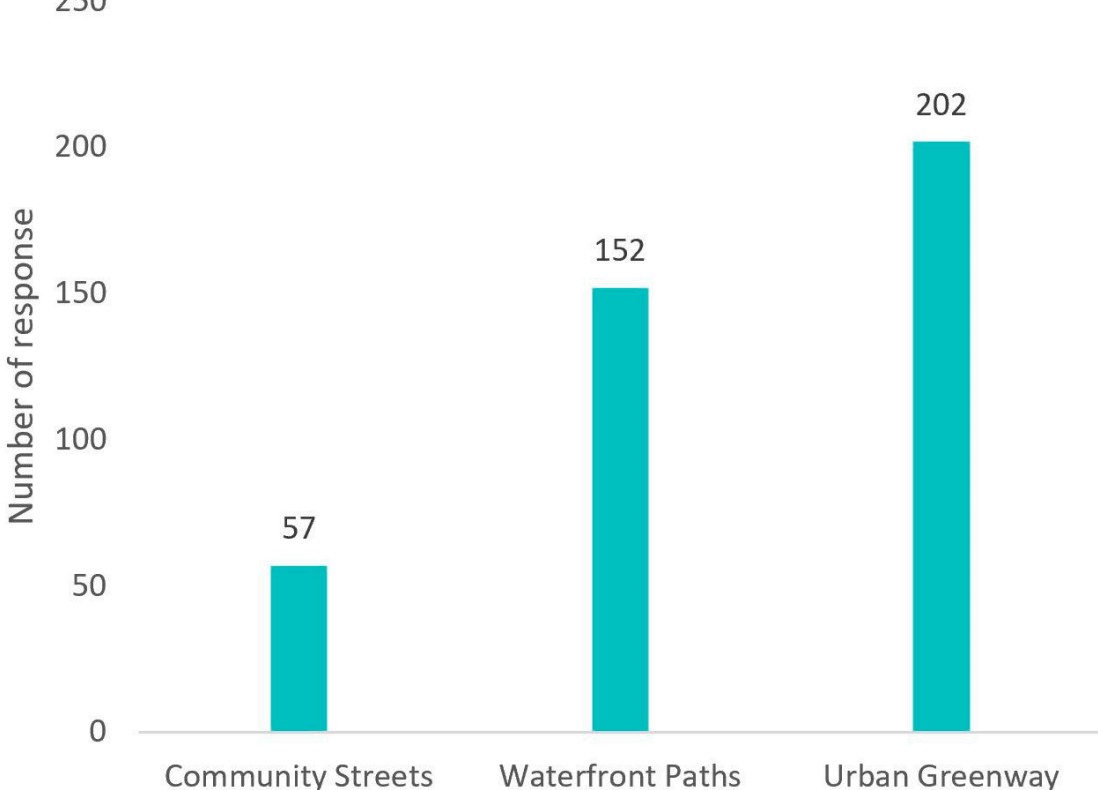

**Figure 3.** Statistical results of responses to three different street settings.

A multinomial logistic regression (with community streets as the reference group) was performed to explore how the respondents' sociodemographic characteristics influenced the choice of their preferred street setting. According to the results shown in Table 3, the model had a good degree of fit ($\chi^2$ = 22.777, $p$ = 0.012). Residents' preferences for street settings were independent of their sociodemographic variables, except for educational level. Respondents with higher educational levels were more likely to prefer waterfront paths than community streets (Exp (B) = 1.715, 95% C. I. 1.141–2.577, $p$ = 0.009).

**Table 3.** Results of multinomial logistic regression of sociodemographic characteristics in three street settings (Community Street as the reference category).

| Street Settings (Ref. Category: Community Streets) | | B | Standard Error | Wald | df | Sig. | Exp (B) | Confidence Interval 95% of Exp (B) | |
|---|---|---|---|---|---|---|---|---|---|
| | | | | | | | | Lower Limit | Upper Limit |
| Waterfront Paths | intercept | 0.691 | 1.273 | 0.295 | 1 | 0.587 | - | | |
| | Age | −0.203 | 0.178 | 1.306 | 1 | 0.253 | 0.816 | 0.576 | 1.156 |
| | Gender | −0.125 | 0.349 | 0.128 | 1 | 0.720 | 0.883 | 0.446 | 1.748 |
| | Education | 0.539 | 0.208 | 6.741 | 1 | 0.009 * | 1.715 | 1.141 | 2.577 |
| | Car Access | 0.365 | 0.410 | 0.793 | 1 | 0.373 | 1.441 | 0.645 | 3.219 |
| | Income | −0.161 | 0.158 | 1.035 | 1 | 0.309 | 0.851 | 0.624 | 1.161 |
| Urban Greenways | intercept | 2.353 | 1.207 | 3.802 | 1 | 0.051 | - | | |
| | Age | −0.263 | 0.170 | 2.397 | 1 | 0.122 | 0.769 | 0.551 | 1.073 |
| | Gender | −0.164 | 0.332 | 0.243 | 1 | 0.622 | 0.849 | 0.443 | 1.628 |
| | Education | 0.160 | 0.197 | 0.660 | 1 | 0.417 | 1.174 | 0.797 | 1.728 |
| | Car Access | 0.141 | 0.391 | 0.130 | 1 | 0.718 | 1.152 | 0.535 | 2.479 |
| | Income | −0.104 | 0.152 | 0.468 | 1 | 0.494 | 0.901 | 0.669 | 1.214 |

AIC = 465.437
BIC = 513.660
Likelihood ratio: $\chi^2$ = 22.777, df = 10, $p$ = 0.012

Note: * $p$ < 0.01. df, degree of freedom. AIC, Akaike information criterion; BIC, Bayesian information criterion.

### 3.2. Physical Environmental Factors Affecting the Choice of Streets for Walking

Table 4 shows an overall assessment of the physical environmental factors. First, the questionnaire was analyzed for reliability, and the results indicated that the five physical environment dimensions were set reliably as all Cronbach's $\alpha$ coefficients exceeded 0.5 [65], ranging from 0.666 to 0.866. According to Landis and Koch (1977), a Cronbach's alpha value greater than 0.8 indicates good internal consistency [66]. Therefore, the transportation facility dimension (Cronbach's alpha = 0.866) and walking conditions dimension (Cronbach's alpha = 0.802) both demonstrated good reliability. A validity test was then conducted, and the results indicated that the current setting of the physical environment dimension was reasonable, and the Kaiser–Meyer–Olkin statistic score was 0.879 ($p < 0.001$).

**Table 4.** Overall assessment of physical environmental factors.

| Dimensions | Items | Differential Analysis on Three Streets | | | Reliability Analysis | Validity Analysis | |
| --- | --- | --- | --- | --- | --- | --- | --- |
| | | Statistics | df | Sig. | Cronbach's Alpha | Kaiser-Meyer-Olkin Statistics | Sig. |
| Functional Facility | Lighting | 1.383 | 2 | 0.501 | 0.666 | 0.879 | <0.001 |
| | Guardrails | 0.802 | 2 | 0.670 | | | |
| | Signage | 3.057 | 2 | 0.217 | | | |
| | Garbage cans | 1.283 | 2 | 0.527 | | | |
| | Wire poles | 4.987 | 2 | 0.083 | | | |
| | Billboards | 1.126 | 2 | 0.569 | | | |
| | Street cameras | 2.684 | 2 | 0.261 | | | |
| | High voltage boxes | 2.122 | 2 | 0.346 | | | |
| Service Facility | Retail stores | 0.552 | 2 | 0.759 | 0.774 | | |
| | Restaurants | 1.332 | 2 | 0.514 | | | |
| | Teahouses | 3.941 | 2 | 0.139 | | | |
| | Bars | 0.296 | 2 | 0.862 | | | |
| | Cafes | 4.558 | 2 | 0.102 | | | |
| | Internet cafes | 0.613 | 2 | 0.736 | | | |
| | Hypermarkets | 4.972 | 2 | 0.083 | | | |
| | Mobile stalls | 2.424 | 2 | 0.298 | | | |
| | Toilets | 0.130 | 2 | 0.937 | | | |
| | Community clinics | 1.577 | 2 | 0.455 | | | |
| | Community fitness equipment | 2.025 | 2 | 0.363 | | | |
| | Community service centers | 2.482 | 2 | 0.289 | | | |
| Transportation Facility | Bus Stations | 4.987 | 2 | 0.083 | 0.866 | | |
| | Subway Stations | 3.196 | 2 | 0.202 | | | |
| | Cab stands | 2.135 | 2 | 0.344 | | | |
| | Bicycle parking | 0.262 | 2 | 0.877 | | | |
| Walking Conditions | Traffic signals | 0.224 | 2 | 0.894 | 0.802 | | |
| | Crossroads | 4.355 | 2 | 0.113 | | | |
| | Ramps | 0.180 | 2 | 0.914 | | | |
| | Sidewalks | 0.443 | 2 | 0.801 | | | |
| | Pedestrian bridges | 1.782 | 2 | 0.410 | | | |
| | Underpasses | 0.753 | 2 | 0.686 | | | |
| | Leveled ground | 4.604 | 2 | 0.100 | | | |
| | Paved Ground | 3.312 | 2 | 0.191 | | | |
| Environmental Quality | Road greenery | 3.248 | 2 | 0.197 | 0.724 | | |
| | Green space along the street | 5.438 | 2 | 0.066 | | | |
| | Artificial landscape | 2.992 | 2 | 0.224 | | | |
| | Litter on the street | 0.259 | 2 | 0.879 | | | |
| | Seating | 0.242 | 2 | 0.886 | | | |
| | Gazebo | 4.987 | 2 | 0.083 | | | |
| | Fountain | 1.990 | 2 | 0.370 | | | |
| | Street tree box with seating | 1.275 | 2 | 0.529 | | | |

Finally, to examine whether the response of residents to physical environmental factors was significantly different between the three street settings, the Kruskal–Wallis H test (with a post hoc test) was executed. Based on the results of the Kruskal–Wallis H test, no significant differences ($p > 0.05$) in response were observed for any of the 40 items from the five physical environment factors, indicating similar response trends among the respondents in three street settings. As there was no significant difference, the impact levels of each item were marked in the follow-up results according to the absolute value of the mean: almost no impact ($|mean| < 1$), moderate impact ($2 > |mean| > 1$), and strong impact ($|mean| > 2$).

3.2.1. Infrastructure

The infrastructure dimension was separated into three sub-dimensions: functional street facilities, service facilities, and transportation facilities (Table 5).

**Table 5.** Residents' responses to infrastructure items for the three street settings.

| Dimensions | Items | Community Streets (N = 57) | | | | Waterfront Paths (N = 152) | | | | Urban Greenways (N = 202) | | | |
|---|---|---|---|---|---|---|---|---|---|---|---|---|---|
| | | M | N | ○ | △ | M | N | ○ | △ | M | N | ○ | △ |
| Functional Street Facility | Lighting | 1.79 | | √ | | 1.82 | | √ | | 1.90 | | √ | |
| | Guardrails | 1.32 | | √ | | 1.07 | | √ | | 1.16 | | √ | |
| | Signage | 1.75 | | √ | | 1.40 | | √ | | 1.45 | | √ | |
| | Garbage cans | 0.11 | √ | | | 0.32 | √ | | | 0.15 | √ | | |
| | Wire poles | −1.23 | | √ | | −0.64 | √ | | | −0.78 | √ | | |
| | Billboards | −0.70 | √ | | | −0.59 | √ | | | −0.53 | √ | | |
| | Street cameras | 0.93 | √ | | | 1.18 | | √ | | 1.00 | | √ | |
| | High voltage boxes | −1.12 | | √ | | −1.06 | | √ | | −1.28 | | √ | |
| Service Facility | Retail stores | 1.35 | | √ | | 1.21 | | √ | | 1.24 | | √ | |
| | Restaurants | 1.04 | | √ | | 0.85 | √ | | | 0.76 | √ | | |
| | Teahouses | 0.02 | √ | | | 0.50 | √ | | | 0.35 | √ | | |
| | Bars | 0.04 | √ | | | −0.08 | √ | | | −0.11 | √ | | |
| | Cafes | −0.23 | √ | | | 0.24 | √ | | | 0.15 | √ | | |
| | Internet cafes | −0.65 | √ | | | −0.42 | √ | | | −0.50 | √ | | |
| | Hypermarkets | 1.42 | | √ | | 0.97 | √ | | | 0.96 | √ | | |
| | Mobile stalls | 0.89 | √ | | | 0.71 | √ | | | 0.45 | √ | | |
| | Toilets | 1.35 | | √ | | 1.20 | | √ | | 1.17 | | √ | |
| | Community clinics | 1.07 | | √ | | 0.76 | √ | | | 0.87 | √ | | |
| | Community fitness equipment | 1.93 | | √ | | 1.74 | | √ | | 1.74 | | √ | |
| | Community service centers | 1.07 | | √ | | 0.96 | √ | | | 1.20 | | √ | |
| Transportation Facility | Bus Stations | 1.33 | | √ | | 0.89 | √ | | | 1.32 | | √ | |
| | Subway Stations | 0.96 | √ | | | 0.80 | √ | | | 1.12 | | √ | |
| | Cab stands | 0.81 | √ | | | 0.59 | √ | | | 0.83 | √ | | |
| | Bicycle parking | 0.96 | √ | | | 0.95 | √ | | | 1.01 | | √ | |

M = Mean value, N = almost no impact, ○ = moderate impact, △ = strong impact.

First, for the functional street facility, the impact of lighting, guardrails, signage, billboards, and high voltage boxes showed similar trends for all three street settings. Specifically, lighting (1.79, 1.82, 1.90), guardrails (1.32, 1.07, 1.16), signage (1.75, 1.4, 1.45), and high-voltage boxes (−1.12, −1.06, −1.28) had a moderate impact on the choice of the street for walking, whereas billboards had a negative but small impact (−0.70, −0.59, −0.53). Notably, wire poles and street cameras had different effects on the three street setting options. Wire poles had a moderately negative impact on community streets (−1.23), whereas this negative impact was negligible for residents who chose waterfront paths (−0.64) and urban greenways (−0.78). Street cameras had almost no impact on the selection of community streets (0.93) but had a moderately positive impact on the selection of the other two street settings (1.18, 1.00).

The impact of retail stores, teahouses, bars, cafes, internet cafes, toilets, and community fitness equipment, on the selection of the three street settings was identical. Specifically, retail stores (1.35, 1.21, 1.24), toilets (1.35, 1.20, 1.17), and community fitness equipment (1.93, 1.74, 1.74) had moderate impacts in all three environments. However, restaurants, hypermarkets, mobile stalls, community clinics, and community service centers had different effects on the selection of the three street settings. For community streets, restaurants (1.04), hypermarkets (1.42), and community clinics (1.07) had moderately positive effects, whereas, for the other two street settings, these service facilities had almost no effect.

Regarding transportation facilities, the cab stands had almost no impact on residents' preferences for the three street settings (0.81, 0.59, 0.83). For community streets and urban greenways, bus stations (1.33, 1.32) had a moderately positive impact, whereas they had almost no impact (0.89) on waterfront paths. Subway stations (1.12) and bicycle parking (1.00) had a moderately positive impact only on urban greenways.

### 3.2.2. Walking Conditions

Table 6 displays the outcomes of the walking condition dimensions. Crossroads, ramps, pedestrian bridges, and underpasses had almost no impact on the selection of these three street settings. In addition, in all three street settings, traffic signals (1.23, 1.1, 1.12), sidewalks (1.25, 1.35, 1.29), and leveled ground (1.98, 1.76, 1.99) had moderately positive impacts. It was also observed that the paved ground dimension had a moderately positive impact on waterfront paths (1.69) and urban greenways (1.66), compared to a strong impact on community streets (2.02).

**Table 6.** Residents' responses to walking condition items for the three street settings.

| Pedestrian Conditions | Community Streets (*N* = 57) | | | | Waterfront Paths (*N* = 152) | | | | Urban Greenways (*N* = 202) | | | |
|---|---|---|---|---|---|---|---|---|---|---|---|---|
| | **M** | **N** | ○ | △ | **M** | **N** | ○ | △ | **M** | **N** | ○ | △ |
| Traffic signals | 1.23 | | √ | | 1.1 | | √ | | 1.12 | | √ | |
| Crossroads | 0.96 | √ | | | 0.63 | √ | | | 0.49 | √ | | |
| Ramps | 0.39 | √ | | | 0.33 | √ | | | 0.41 | √ | | |
| Sidewalks | 1.25 | | √ | | 1.35 | | √ | | 1.29 | | √ | |
| Pedestrian bridges | 0.63 | √ | | | 0.67 | √ | | | 0.90 | √ | | |
| Underpasses | 0.63 | √ | | | 0.49 | √ | | | 0.47 | √ | | |
| Leveled ground | 1.98 | | √ | | 1.76 | | √ | | 1.99 | | √ | |
| Paved ground | 2.02 | | | √ | 1.69 | | √ | | 1.66 | | √ | |

M = Mean value, N = almost no impact, ○ = moderate impact, △ = strong impact.

### 3.2.3. Environmental Quality

Table 7 shows the results for the environmental quality dimension. There was a moderately positive impact of gazebos (1.63, 1.71, 1.97), fountains (1.88, 1.64, 1.65), and street tree boxes with seating (1.82, 1.82, 1.95) on the choice of all three street settings. For both community streets and urban greenways, seating (2.04, 2.02) had a strong positive impact, indicating that the presence of seating was crucial to residents in determining the choice to walk on these two types of streets. In addition, for waterfront paths and urban greenways, road greenery (2.20, 2.21), green space along the street (2.22, 2.13), and artificial landscapes (2.06, 2.00) had a strong positive impact, indicating that residents strongly preferred both paths because of the presence of green and artificial elements. Not surprisingly, litter had a non-negligible negative impact on any type of street choice (−1.7, −1.72, −1.59).

### 3.3. Aesthetic Environmental Factors Affecting the Choice of Streets for Walking

Table 8 shows the overall assessment of aesthetic environmental factors. The questionnaire reliability analysis results indicated that the two aesthetic environment dimensions (Subjective Perception and Sensory Perception) were set reliably, as all Cronbach's α coefficients ranged from 0.573 to 0.831. The validity test indicated that the current setting of the aesthetic environment dimension was reasonable, and the Kaiser–Meyer–Olkin statistic

was 0.882 ($p < 0.001$). Similar to the physical environmental factors and, based on the results of the Kruskal–Wallis H test, no significant differences ($p > 0.05$) in responses were observed for any of the nine items from the two aesthetic environmental factors, indicating similar response trends among the respondents in the three street settings. As there was no significant difference, the impact levels of each item were marked in the follow-up results according to the absolute value of the mean: almost no impact (|mean| < 1), moderate impact (2 > |mean| > 1), and strong impact (|mean| > 2).

**Table 7.** Residents' responses to environmental quality items for the three street settings.

| Pedestrian Conditions | Community Streets (N = 57) | | | | Waterfront Paths (N = 152) | | | | Urban Greenways (N = 202) | | | |
|---|---|---|---|---|---|---|---|---|---|---|---|---|
| | **M** | **N** | ○ | △ | **M** | **N** | ○ | △ | **M** | **N** | ○ | △ |
| Road greenery | 1.98 | | √ | | 2.20 | | | √ | 2.21 | | | √ |
| Green space along the street | 1.88 | | √ | | 2.22 | | | √ | 2.13 | | | √ |
| Artificial landscape | 1.77 | | √ | | 2.06 | | | √ | 2.00 | | | √ |
| Litter on the street | −1.7 | | √ | | −1.72 | | √ | | −1.59 | | √ | |
| Seating | 2.04 | | | √ | 1.93 | | √ | | 2.02 | | | √ |
| Gazebo | 1.63 | | √ | | 1.71 | | √ | | 1.97 | | √ | |
| Fountain | 1.88 | | √ | | 1.64 | | √ | | 1.65 | | √ | |
| Street tree box with seating | 1.82 | | √ | | 1.82 | | √ | | 1.95 | | √ | |

M = Mean value, N = almost no impact, ○ = moderate impact, △ = strong impact.

**Table 8.** Overall assessment of aesthetic environmental factors.

| Dimensions | Items | Differential Analysis on Three Streets | | | Reliability Analysis | Validity Analysis | |
|---|---|---|---|---|---|---|---|
| | | **Statistics** | **df** | **Sig.** | **Cronbach's Alpha** | **Kaiser-Meyer-Olkin Statistics** | **Sig.** |
| Subjective Perception | Diversity | 0.499 | 2 | 0.779 | 0.831 | 0.882 | <0.001 |
| | Facility Accessibility | 0.034 | 2 | 0.983 | | | |
| | Uniformity | 4.213 | 2 | 0.122 | | | |
| | Novelty | 0.368 | 2 | 0.832 | | | |
| | Maintainability | 3.498 | 2 | 0.174 | | | |
| | Charm | 2.512 | 2 | 0.285 | | | |
| | Sense of mystery | 3.000 | 2 | 0.223 | | | |
| Sensory Perception | Pleasant sounds | 1.895 | 2 | 0.388 | 0.573 | | |
| | Pleasant odors | 0.634 | 2 | 0.728 | | | |

The results of the impact of aesthetic quality are listed in Table 9. The diversity (1.65, 1.63, 1.71), facility accessibility (1.79, 1.80, 1.83), uniformity (1.63, 1.46, 1.72), novelty (1.65, 1.61, 1.69), and sense of mystery (1.77, 1.67, 1.52) dimensions had positive moderate impacts on the three street setting choices. Notably, the maintainability dimension (2.10) had a strong positive impact on the choice of urban greenways. For community streets and urban greenways, charm dimensions (2.02, 2.00) had a strong positive impact, whereas they had a moderate positive impact on waterfront paths (1.87). In addition, pleasant odors (2.01) had a strong positive impact on waterfront paths but a moderate impact on the other two settings (1.86, 1.97). The pleasant sounds dimension (1.13, 1.41, 1.25) also had a moderate positive impact on all three streets.

**Table 9.** Residents' responses to aesthetic environmental quality items for the three street settings.

| Dimensions | Items | Community Streets (N = 57) | | | | Waterfront Paths (N = 152) | | | | Urban Greenways (N = 202) | | | |
|---|---|---|---|---|---|---|---|---|---|---|---|---|---|
| | | **M** | **N** | ○ | △ | **M** | **N** | ○ | △ | **M** | **N** | ○ | △ |
| Subjective perception | Diversity | 1.65 | | √ | | 1.63 | | √ | | 1.71 | | √ | |
| | Facility Accessibility | 1.79 | | √ | | 1.8 | | √ | | 1.83 | | √ | |
| | Uniformity | 1.63 | | √ | | 1.46 | | √ | | 1.72 | | √ | |
| | Novelty | 1.65 | | √ | | 1.61 | | √ | | 1.69 | | √ | |
| | Maintainability | 1.89 | | √ | | 1.94 | | √ | | 2.10 | | | √ |
| | Charm | 2.02 | | | √ | 1.87 | | √ | | 2.00 | | | √ |
| | Sense of mystery | 1.77 | | √ | | 1.67 | | √ | | 1.52 | | √ | |
| Sensory perception | Pleasant sounds | 1.13 | | √ | | 1.41 | | √ | | 1.25 | | √ | |
| | Pleasant odors | 1.86 | | √ | | 2.01 | | | √ | 1.97 | | √ | |

M = Mean value, N = almost no impact, ○ = moderate impact, △ = strong impact.

## 4. Discussion

### 4.1. Sociodemographic Characteristic's Impact on Residents' Preference to Walking Street

The frequencies of residents' preferred walking street settings (community streets, waterfront paths, and urban greenways) were tallied separately, and a multinomial logistic regression analysis was used to explore how respondents' sociodemographic characteristics influenced the choice of their preferred street setting. The above test revealed that residents' preference for street settings were independent of sociodemographic variables, except for their educational level. Specifically, respondents with higher educational levels were more likely to prefer streets with water elements (or blue spaces) than community streets (pure built environment). This is a valuable finding; to the best of our knowledge, few studies have addressed the association between street environment preferences and demographic characteristics. Thus, our findings answer the first question posed in this study and have implications for the development of future outdoor exercise interventions in cities.

### 4.2. Impact of Physical Environmental Dimensions on the Three Street Settings

According to the results, we determined that the physical environmental dimensions (infrastructure, walking conditions, and environmental quality) had an impact on residents' choices and that these impacts exhibited differences across street settings.

Community streets, restaurants, hypermarkets, community clinics, and community service centers had a greater impact, and paved streets had a stronger impact. A residential street with favorable physical environmental factors encouraged residents to walk [67,68]. Thus, it is likely that residents who prefer to walk on community streets are more concerned with the infrastructure on those streets. Consequently, infrastructure, especially functional street facilities, has a greater impact on their choice.

For the waterfront path, road greenery, green spaces along the street, and artificial landscapes had a greater impact. However, street infrastructure had a smaller impact. This was likely because residents who prefer to walk on waterfront paths are more concerned with the natural landscape, green space, and vegetation, which all impact the waterfront's natural environment [69]. In addition, residents who prefer waterfront paths are more concerned with natural environmental quality; consequently, environmental quality has a greater impact.

Road greenery, green spaces along the street, artificial landscapes, and seating had a strong impact on urban greenways. Moreover, the transportation facility dimensions in the infrastructure, such as bus stations, subway stations, and bicycle parking, had a greater impact on residents' walking on the greenway than the other two paths. This is likely because green spaces can provide health benefits and have a positive impact on residents [20,27]; consequently, residents subconsciously associate a higher degree of the green environment with positive benefits. Furthermore, urban greenways are often located far from residential areas (such as at the edge of cities), which results in higher

transportation facility needs. Thus, environmental quality and transportation facilities had a greater impact on residents who chose urban greenways.

### 4.3. Impact of Aesthetic Environmental Dimensions on the Three Street Settings

Based on these findings, we determined that the aesthetic quality of the environment had a positive and moderate impact on the residents of all three environments and that the overall aesthetic quality of the environment had a greater impact on residents than their physical environment. Street maintenance had a stronger impact on residents who chose urban greenways in terms of the impact of the subjective perception of residents' street aesthetics. Improving or maintaining the quality of the built environment may increase residents' walking activities [67].

Hence, urban greenways should prioritize the maintenance of streets, including the maintenance of street infrastructure and green spaces. In addition, the charm has a stronger impact on community streets and urban green spaces; consequently, the design or renovation of community streets and urban greenways should prioritize attracting more residents based on contemporary everyday aesthetic concepts [70].

The sensory perception of a street can have an impact on residents' choice of all three paths, as they prefer to hear pleasant sounds from nature over city noise. Therefore, designs should consider residents' senses of sound and odor [71]. Notably, odors have a strong impact on residents who choose waterfront paths; therefore, plants that release pleasant odors should be used to design or renovate waterfront paths.

### 4.4. Study Significance and Implications

According to this study, the physical and aesthetic qualities of the street environment can directly influence residents walking preferences. Consequently, through the design of the physical and aesthetic qualities of different types of street environments, it is possible to increase their use by residents. This can potentially increase their walking time and frequency and promote health. Based on the results of this survey, three environmental factors in different street settings had a non-negligible impact on residents' selection of preferred walking streets and are summarized in Table 10.

**Table 10.** Residents' responses for preferred walking environment items for the three street settings.

| Factors | Dimensions | Community Streets | | Waterfront Paths | | Urban Greenways | |
|---|---|---|---|---|---|---|---|
| | | **Moderate Impact** | **Strong Impact** | **Moderate impact** | **Strong Impact** | **Moderate Impact** | **Strong Impact** |
| Physical environmental factors | Infrastructure | Lighting Guardrails Signage Wire poles High-voltage boxes Retail stores Restaurants Hypermarkets Toilets Community clinics Community fitness equipment Community service centers Bus stations | | Lighting Guardrails Signage Street cameras High voltage boxes Retail stores Toilets Community fitness equipment | | Lighting Guardrails Signage Street cameras High voltage boxes Retail stores Toilets Community fitness equipment Community service canters Bus stations Subway stations Bicycle parking | |

**Table 10.** *Cont.*

| Factors | Dimensions | Community Streets | | Waterfront Paths | | Urban Greenways | |
|---|---|---|---|---|---|---|---|
| | | Moderate Impact | Strong Impact | Moderate impact | Strong Impact | Moderate Impact | Strong Impact |
| | Pedestrian conditions | Traffic signals Sidewalk Leveled ground | Paved ground | Traffic signals Sidewalk Leveled ground Paved ground | | Traffic signal sidewalk Leveled ground Paved ground | |
| | Environmental quality | Road greening Green spaces along the street Artificial landscape Litter on the street Gazebo Fountain street tree box with seating | Seating | Litter on the street Seating Gazebo Fountain street tree box with seating | Road greening Green spaces along the street Artificial landscape | Litter on the street Gazebo Fountains street tree box with seats | Road greening Green spaces along the street Artificial landscape Seating |
| Aesthetic environmental factors | Subjective perception | Diversity Facility Accessibility Uniformity Novelty Maintainability Sense of mystery | Charm | Diversity Facility Accessibility Uniformity Novelty Maintainability Charm Sense of mystery | | Diversity Facility Accessibility Uniformity Novelty Fascination Sense of mystery | Maintainability Charm |
| | Sensory perception | Pleasant sounds Pleasant odors | | Pleasant sounds | Pleasant odors | Pleasant sounds Pleasant odors | |

Based on this study, design solutions for the three distinct street environments were proposed. First, because residents are more concerned with the physical environment of community streets (Figure 4), infrastructure on community streets should be well-developed, including lighting, guardrails, seating, and paved ground. However, it has been discovered that high-voltage boxes and wire poles have a negative impact on residents, and street designers should conceal these facilities for the sake of aesthetics. In addition, the findings show that residents are also interested in restaurants, hypermarkets, and community clinics on community streets. Hence, these facilities are more likely to encourage residents to walk on community streets. Residents of community streets are also interested in restrooms and exercise equipment; therefore, there should be an abundance of street facilities for walking and resting.

Second, in the design of a waterfront path environment (Figure 5), factors such as street cameras, seating, gazebos, fountains, signage, and artificial landscapes should be incorporated. In addition, the natural environment plays a significant role in encouraging residents to walk, so physical environmental facilities should be integrated into the natural environment to create a natural recreational public space, as residents prefer the environment's natural state. When establishing street facilities, road greenery, green areas along the street, and plants that emit pleasant odors should be considered, and residents should be provided with sufficient space for walking and other activities.

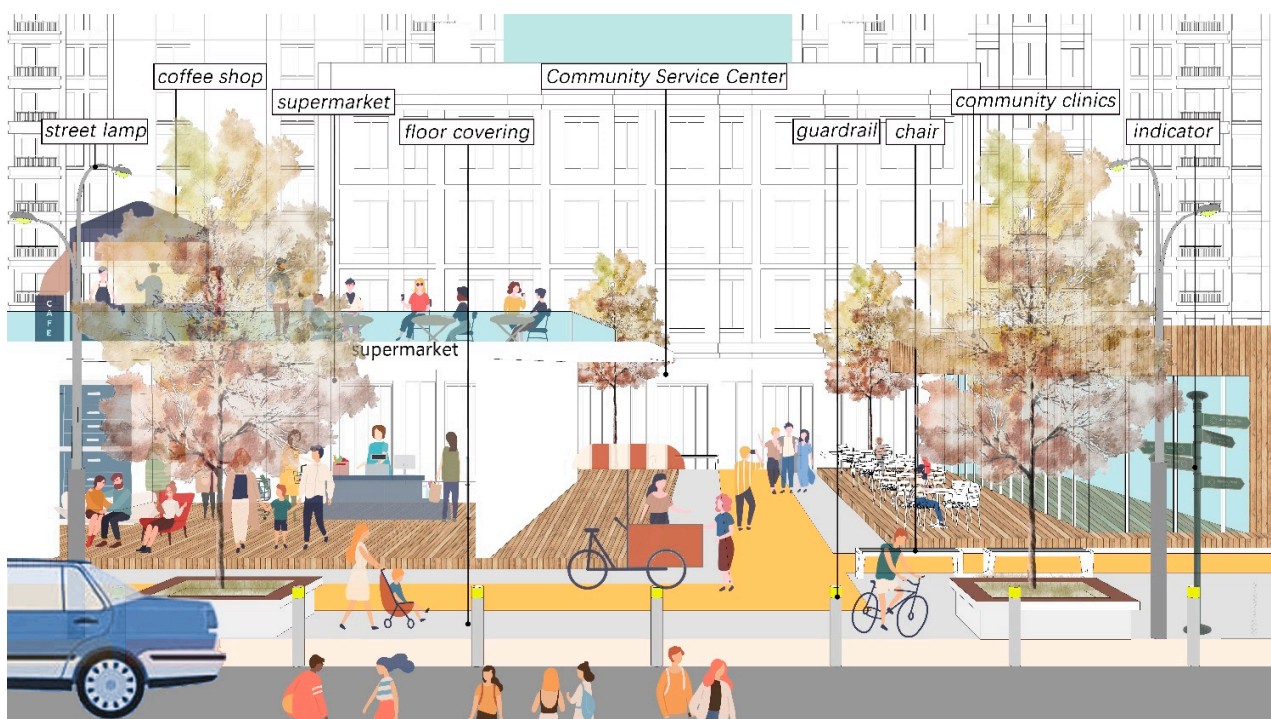

**Figure 4.** Community street design scenes.

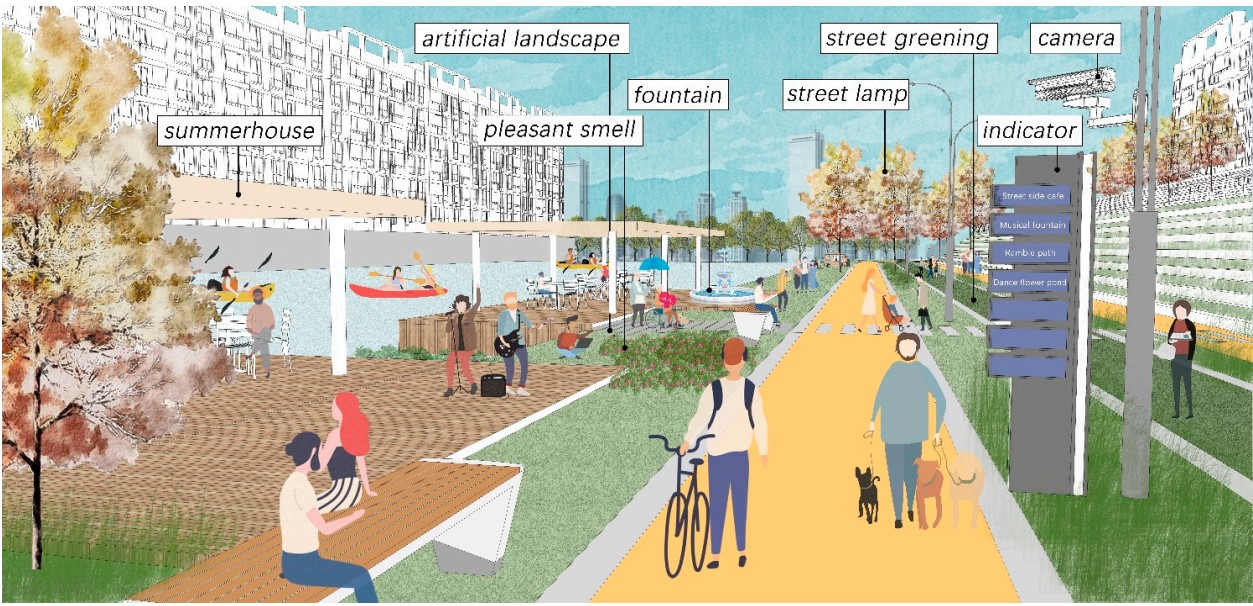

**Figure 5.** Waterfront path design scenes.

Third, infrastructure such as community service centers, subway stations, bicycle parking, seating, and artificial landscapes in an urban greenway environment (Figure 6) is worth considering. Additionally, the presence of road greenery and green spaces along the street is a significant factor that encourages residents to walk. Therefore, green spaces on urban greenways should satisfy the walking needs of residents, allowing them to experience the natural environment of urban greenways and provide pedestrians with green spaces for rest and recreation. Furthermore, streets should be well maintained to appeal to residents and encourage them to walk.

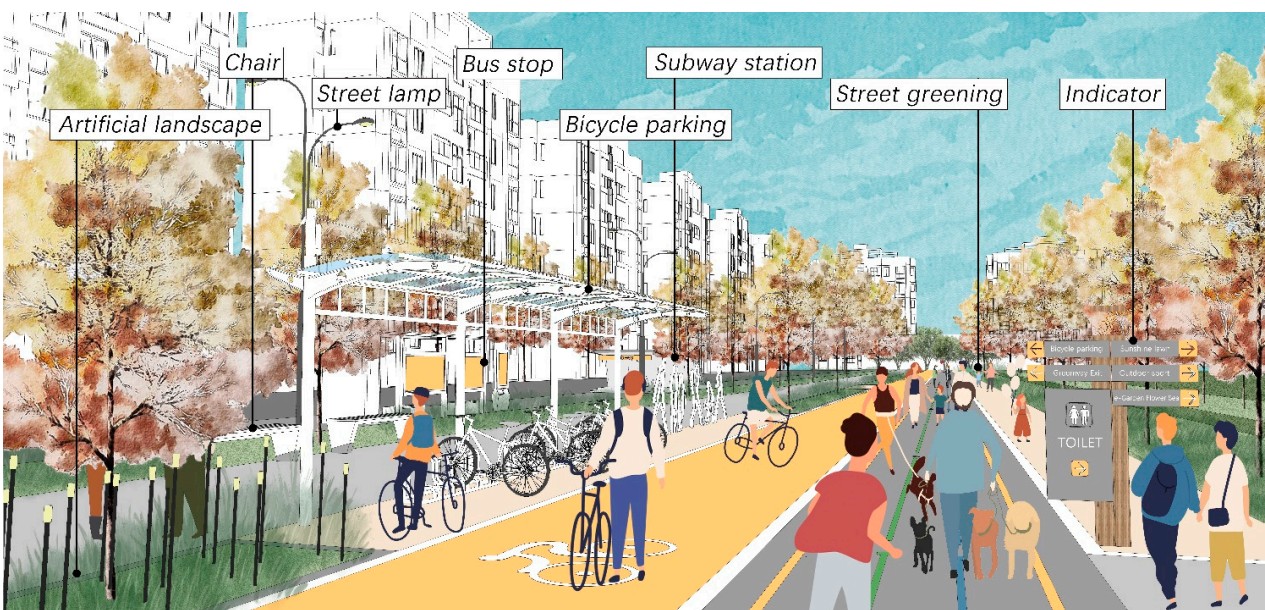

**Figure 6.** Urban greenway design scenes.

## 5. Conclusions

These three different types of street environments have different impacts on the walking behavior of residents. A well-designed or renovated street environment can boost the walking time and frequency of residents in different street environments. Therefore, a street environment is essential for establishing a healthy environment for residents. This study examined the impact of street environmental factors on residents in three street environments (community streets, waterfront paths, and urban greenways) using a questionnaire survey for 411 Chengdu metropolitan residents. Current research has found that: (1) people with higher levels of education prefer streets with water bodies as walking paths; (2) the environmental quality of physical and aesthetic aspects has an impact on residents' choices, and the aesthetic environment quality has a stronger impact; (3) the impact of most infrastructure items on community streets are stronger than on other streets; (4) residents are more concerned about the environmental quality of waterfront paths and urban greenways. In addition, the design inclinations of urban planners vary depending on the street environment. Community streets should emphasize street infrastructure and provide adequate walking conditions and environmental quality for pedestrians. Waterfront paths should emphasize streetscapes and pedestrian-friendly roads to ensure there is sufficient space for residents to relax and enjoy recreation. Urban greenways should offer sufficient green space for city residents to experience the city's greenery.

In summary, this study used a bottom-up approach to identify the varying impacts of environmental dimensions on residents walking in three distinct environmental streets and, in doing so, assists future policymakers and urban planners in implementing various street designs or renovations. Consequently, different types of urban streets can be designed or renovated based on these findings. They can be modified to incorporate different physical environmental factors, as well as aesthetic environmental factors, to attract residents and encourage more walking activities to improve physical and mental health.

**Author Contributions:** Conceptualization, Q.Y., S.L. and J.J.; methodology, Q.Y. and S.L.; software, Q.Y., S.L. and J.J.; validation, Q.Y. and S.L.; formal analysis, Q.Y. and S.L.; resources, Q.Y., S.L. and J.J.; data curation, Q.Y., S.L. and J.J.; writing—original draft preparation, Q.Y. and S.L.; writing—review and editing, Q.Y. and S.L.; visualization, Q.Y.; supervision, Q.Y., S.L. and J.J.; funding acquisition, Q.Y. All authors have read and agreed to the published version of the manuscript.

**Funding:** This project is supported by Jiangsu Funding Program for Excellent Postdoctoral Talent (2022ZB599) and the special fund of Collaborative Innovation Center of Soochow University and Suzhou Yuanke Ecological Construction Group (SY2022003).

**Data Availability Statement:** The data that support the findings of this study are available from the first author Q.Y. and corresponding author, S.L., upon reasonable request.

**Acknowledgments:** We would like to thank the reviewers and editors for their constructive comments that helped us improve the quality of the manuscript. We appreciate the valuable information provided by all the anonymous interviewees.

**Conflicts of Interest:** The authors declare no conflict of interest.

## Appendix A

Survey of the impact of street facilities and environment on pedestrians

1. Age [Single-choice]

   ( ) 18–25 years old
   ( ) 26–55 years old
   ( ) 55 years old or older

2. Gender [Single-choice]

   ( ) Male
   ( ) Female

3. Occupation [Single-choice]

   ( ) Medical personnel (doctors, nurses)
   ( ) Teachers, lawyers, service industry workers (caterers/drivers/salesmen, etc.)
   ( ) Freelancer (e.g., writer/artist/photographer/guide, etc.)
   ( ) Workers (e.g., factory workers/construction workers/city sanitation workers, etc.)
   ( ) Company employees
   ( ) Career/civil servants/government workers
   ( ) Students
   ( ) Housewife
   ( ) No job/retired

4. Educational level [Single-choice]

   ( ) High school and below
   ( ) College
   ( ) Undergraduate
   ( ) Graduate

5. Whether or not to hold a driver's license [radio]

   ( ) Yes
   ( ) No

6. Your monthly income [Single-choice]

   ( ) Less than 1500 RMB
   ( ) 1500–3500 RMB
   ( ) 3500–4500 RMB
   ( ) 4500–8000 RMB
   ( ) 8000–15,000 RMB
   ( ) Above 15,000 RMB

7.   Preferred environment for walking [Single-choice]

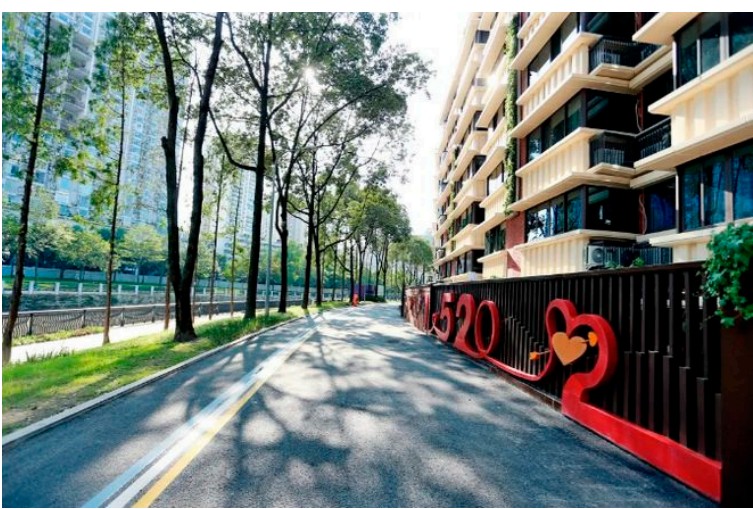

(    ) Community Street

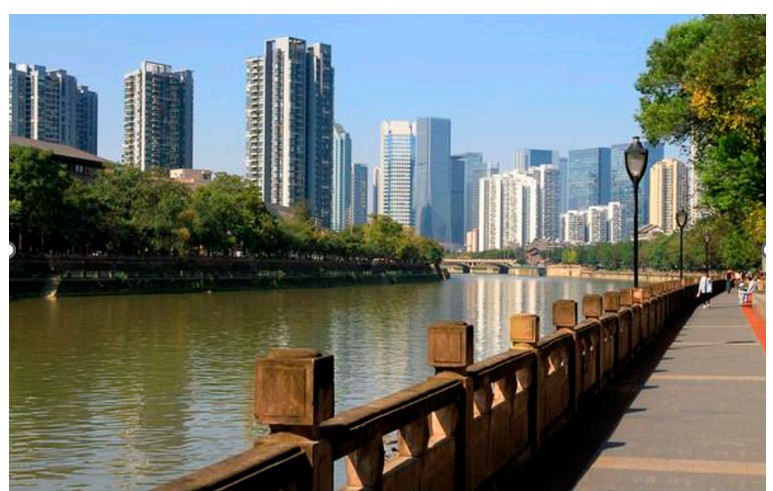

(    ) Waterfront path

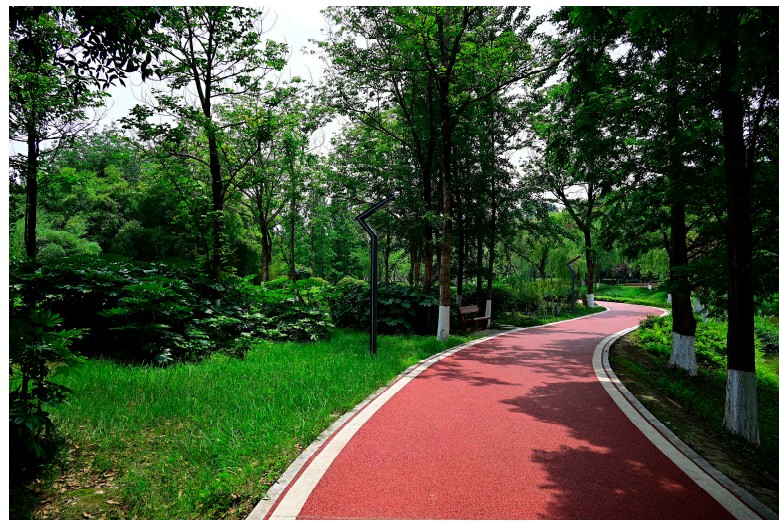

(    ) Urban Greenway

8. How do you think the presence of infrastructure on streets physical environmental factors of infrastructure affects your choice of that road as a walking path? (A higher negative value means less likely to choose the path, a higher positive value means more likely to choose the path, 0 means no effect) [Matrix scale question]

**Table A1.** Preferences for streets physical environmental factors of infrastructure.

| Environmental Factors | Score | | | | | | |
|:---:|:---:|:---:|:---:|:---:|:---:|:---:|:---:|
| | −3 | −2 | −1 | 0 | 1 | 2 | 3 |
| Streetlights and lighting facilities | | | | | | | |
| Guardrails | | | | | | | |
| Signage | | | | | | | |
| Garbage cans | | | | | | | |
| Wire poles | | | | | | | |
| Billboards | | | | | | | |
| Street cameras | | | | | | | |
| High voltage boxes | | | | | | | |
| Retail stores | | | | | | | |
| Restaurants | | | | | | | |
| Teahouses | | | | | | | |
| Bars | | | | | | | |
| Cafés | | | | | | | |
| Internet cafes | | | | | | | |
| Food markets/hypermarkets | | | | | | | |
| Mobile stalls | | | | | | | |
| Toilets | | | | | | | |
| Community clinics/hospitals | | | | | | | |
| Community fitness equipment | | | | | | | |
| Community service centers | | | | | | | |
| Bus Stations | | | | | | | |
| Subway Stations | | | | | | | |
| Cab stands | | | | | | | |
| Bicycle parking | | | | | | | |

9. How do you think the presence of streets physical environmental factors of walking conditions affects your choice of that road as a walking path? (A higher negative value means less likely to choose the path, a higher positive value means more likely to choose the path, 0 means no effect) [Matrix scale question]

**Table A2.** Preferences for streets physical environmental factors of walking conditions.

| Environmental Factors | Score | | | | | | |
|:---:|:---:|:---:|:---:|:---:|:---:|:---:|:---:|
| | −3 | −2 | −1 | 0 | 1 | 2 | 3 |
| Traffic signals | | | | | | | |
| Crossroads | | | | | | | |
| Ramps | | | | | | | |
| Sidewalks | | | | | | | |
| Pedestrian bridges | | | | | | | |
| Underpasses | | | | | | | |
| Level ground | | | | | | | |
| Ground paving | | | | | | | |
| Road greenery | | | | | | | |

10. How do you think the presence of streets physical environmental factors of environmental quality affects your choice of that road as a walking path? (A higher negative value means less likely to choose the path, a higher positive value means more likely to choose the path, 0 means no effect) [Matrix scale question]

**Table A3.** Preferences for streets physical environmental factors of environmental quality.

| Environmental Factors | Score | | | | | | |
|---|---|---|---|---|---|---|---|
| | −3 | −2 | −1 | 0 | 1 | 2 | 3 |
| Road greenery | | | | | | | |
| Green space along the street | | | | | | | |
| Artificial landscape | | | | | | | |
| Litter on the street | | | | | | | |
| Seating | | | | | | | |
| Gazebo | | | | | | | |
| Fountain | | | | | | | |
| Tree pond with seating | | | | | | | |

11. How do you think the presence of the streets s aesthetic environmental factors affects your choice of that road as a walking path? (A higher negative value means less likely to choose the path, a higher positive value means more likely to choose the path, 0 means no effect) [Matrix Scale Questions]

**Table A4.** Preferences for streets aesthetic environmental factors.

| Dimension | Environmental Factors | Score | | | | | | |
|---|---|---|---|---|---|---|---|---|
| | | −3 | −2 | −1 | 0 | 1 | 2 | 3 |
| Subjective perception | Diversity | | | | | | | |
| | Convenience | | | | | | | |
| | Uniformity | | | | | | | |
| | Novelty | | | | | | | |
| | Maintainability | | | | | | | |
| | Charm | | | | | | | |
| | Sense of mystery | | | | | | | |
| Sensory perception | Pleasant sounds | | | | | | | |
| | Noise | | | | | | | |
| | Pleasant odors | | | | | | | |

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
