# Peer review of "Urban Residents’ Preferred Walking Street Setting and Environmental Factors: The Case of Chengdu City"

_buildings, doi:10.3390/buildings13051199_

Round 1
Reviewer 1 Report
Dear authors,
Firstly, I would like to thank you for your work in this field.
You did a good job!
After a comprehensive reading, I only have a few comments for the current version, which are not many but essential.
For revision details, please see the followings:
[1] I suggest you add more keywords, it can help you improve the exposure.
[2] I suggest you add Chengdu in your introduction, as the case study, mentioned it in your introduction is reasonable.
[3] I suggest you cite the following reference. Your research field is similar.
* Li, Y., Peng, L., Wu, C., & Zhang, J. (2022). Street view imagery (svi) in the built environment: A theoretical and systematic review. Buildings, 12(8), 1167.
[4] I suggest you re-consider figure 1, because each district border is blurry for the municipal distribution. Meanwhile, the location of the Jinjiang district has a little problem because I think it is Shuangliu?
[5] for the questionnaire figure, I suggest replacing them with more high-resolution figures.
Again, I have a high comment on your contribution.
The upon-revision recommendations are helping this manuscript be more readable for readers.
bbgvb
Author Response
We would like to thank you very much for your careful and thorough reading of this manuscript and for the thoughtful comments and constructive suggestions, which help to improve the quality of this manuscript. Here is a point-by-point response to your comments and concerns. All page and line numbers refer to the revised manuscript file.
OVERALL IMPRESSION
You did a good job! After a comprehensive reading, I only have a few comments for the current version, which are not many but essential.
Point 1: - Keywords, line 21
I suggest you add more keywords, it can help you improve the exposure.
Response 1: Thank you for this helpful comment. We have added some additional keywords, including Physical environmental factors, Aesthetic environmental factors and Online questionnaire.
Point 2: - Introduction
I suggest you add Chengdu in your introduction, as the case study, mentioned it in your introduction is reasonable.
Response 2: We strongly agree with your comments, however, since we have already introduced Chengdu in the Research Area section, we have not inserted an introduction to Chengdu in the Introduction section to avoid repetition (lines 133-142).
Point 3: - Introduction
I suggest you cite the following reference. Your research field is similar.
Response 3: Thanks for this comment, I have added this helpful paper in our Introduction section.
Point 4: - Materials and Methods
I suggest you re-consider figure 1, because each district border is blurry for the municipal distribution. Meanwhile, the location of the Jinjiang district has a little problem because I think it is Shuangliu?
Response 4: Many thanks for your careful examination of our manuscript. We have modified the figure 1 so that the reader could read the correct district border very clearly.
Point 5: - Appendix
for the questionnaire figure, I suggest replacing them with more high-resolution figures.
Response 5: Thank you for pointing this out. We have changed our questionnaire figure into a high-resolution one.
Reviewer 2 Report
No comments
Author Response
We would like to thank you for your review of our manuscript. We have revised the manuscript based on two other reviewers.
Reviewer 3 Report
Dear authors,
Thanks for the interesting article. Despite the fact that the results of the article are valuable, the research methodology is very simple. However, I have found several aspects that require your attention:
1) line 32, remove the dot.
2) line 212, it's not clear what age groups you're talking about until I look at table 2, which is on another page.
3) line 218-219, typo, when you indicate 202 respondents, you are talking about urban greenway, and not about community streets again.
4) but the most important question is: why within the profile of respondents, physical and aesthetic environments, you use a specific list of factors, not some other, or wider or less, but this one. For example, why is there no such factor as "married or single" or "have children or no children" among the profile of respondents? I am sure that with the factor about the presence of children there will definitely be a correlation with preferences about the design of recreational areas, spaces for walking, etc.
Best wishes,
reviewer
Author Response
We would like to thank you for the careful and thorough reading of this manuscript and for the thoughtful comments and constructive suggestions, which help to improve the quality of this manuscript. Here is a point-by-point response to your comments and concerns. All page and line numbers refer to the revised manuscript file.
OVERALL IMPRESSION
Thanks for the interesting article. Despite the fact that the results of the article are valuable, the research methodology is very simple. However, I have found several aspects that require your attention:
Point 1:
line 32, remove the dot.
Response 1: Thank you for pointing this out, and now the dot has been removed (line 33).
Point 2: - Materials and Methods
line 212, it's not clear what age groups you're talking about until I look at table 2, which is on another page.
Response 2: Many thanks for your careful examination of our manuscript. We have modified the unclear sentence.
The detail as follow:
(Line 217-219) In addition, the number of respondents was similar in all three age groups (18-25, 26-55, and over 55 years old).
Point 3: - Introduction
line 218-219, typo, when you indicate 202 respondents, you are talking about urban greenway, and not about community streets again.
Response 3: Thank you for pointing this out. We have revised the mistake.
Point 4: - Materials and Methods
but the most important question is: why within the profile of respondents, physical and aesthetic environments, you use a specific list of factors, not some other, or wider or less, but this one. For example, why is there no such factor as "married or single" or "have children or no children" among the profile of respondents? I am sure that with the factor about the presence of children there will definitely be a correlation with preferences about the design of recreational areas, spaces for walking, etc.
Response 4: Thanks for this insightful comment. First, for the physical and aesthetic environmental factors, we referred to many previous studies. However, it was difficult to cover all the factors, so we selected only the factors that we considered most important, and please refer to the literature in Table 1 for details. Second, as this comment states, we only surveyed a limited number of socio-demographic characteristics, and we did not survey potentially useful items such as "presence of children". We strongly agree with this useful comment, so we will cover more socio-demographic characteristics in future surveys.